# Nutritional Intervention Effectiveness on Slowing Time to Dialysis in Elderly Patients with Chronic Kidney Disease—A Retrospective Cohort Study

**DOI:** 10.3390/geriatrics7040083

**Published:** 2022-08-13

**Authors:** Simone Perna, Fatima Faisal, Daniele Spadaccini, Tariq A. Alalwan, Zahra Ilyas, Clara Gasparri, Mariangela Rondanelli

**Affiliations:** 1Department of Biology, College of Science, Sakhir Campus, University of Bahrain, Zallaq 32038, Bahrain; 2Department of Health Sciences, University of Piemonte Orientale, 28100 Novara, Italy; 3Endocrinology and Nutrition Unit, Azienda di Servizi alla Persona “Istituto Santa Margherita”, University of Pavia, 27100 Pavia, Italy; 4IRCCS Mondino Foundation, 27100 Pavia, Italy; 5Department of Public Health, Experimental and Forensic Medicine, University of Pavia, 27100 Pavia, Italy

**Keywords:** chronic kidney disease, nutritional intervention, dialysis, kidney, glomerular filtration rate

## Abstract

Background: Chronic kidney disease (CKD) is a global health issue. Evidence of the effectiveness of nutritional intervention on slowing time to dialysis is limited in Arab countries. Therefore, this study aims to contribute to current research by providing new insights on the efficacy of personalized nutritional intervention in pre-dialysis patients in the Kingdom of Bahrain. Methods: This retrospective cohort study included 265 CKD patients (163 males and 108 females) who were admitted to the nephrology outpatient clinic at Salmaniya Medical Complex in Bahrain. The nutritional intervention group (NIG) receiving dietary advice by an expert renal dietitian consisted of 121 patients, while the non-nutritional intervention control group (non-NIG) that did not receive any nutritional support consisted of 150 patients. Patients were evaluated at baseline and follow-up. Results: The NIG had a significant increase in the estimated glomerular filtration rate (eGFR) compared to the non-NIG that had a significant decrease (5.16 vs. −2.85 mL/min/1.73 m^2^ (*p* = 0.000), respectively). When adjusted for age and gender, the mean difference was greater (8.0 mL/min/1.73 m^2^, *p* = 0.000). Additionally, there was a significant reduction in blood urea nitrogen and serum creatinine (−2.19 mmol/L and −25.31 µmol/L; *p* = 0.000, respectively). Moreover, the intervention had a positive impact on weight loss and body mass index (−1.84 kg and−0.69 kg/m^2^, respectively; *p* = 0.000) and lipid profile, with a significant reduction in total cholesterol and triglyceride levels (−0.17 mmol/L, *p* = 0.006 and −0.15 mmol/L, *p* = 0.026, respectively). Additional significant results from the NIG included reduced uric acid (−28.35 µmol/L, *p* = 0.006), serum phosphorus (−0.05 mol/L, *p* = 0.025), fasting blood glucose (−0.70 mmol/L, *p* = 0.016) and glycated hemoglobin (1.10 mmol/mol, *p* = 0.419). Conclusions: This study suggests that, in patients of CKD, nutritional intervention counselling plays a significant role in reducing the time needed for dialysis and improves nutritional-related biomarkers compared to patients not receiving this intervention.

## 1. Introduction

The global health burden of chronic kidney disease (CKD) is increasing rapidly, affecting 850 million people worldwide and is predicted to become the fifth most common leading cause of death worldwide by 2040 [1]. In the Kingdom of Bahrain, the prevalence rate of CKD increased by 10% yearly with almost a two-third (64%) increase during the 2010–2020 decade.

CKD management requires a multidisciplinary approach with involvement of highly specialized team members such as nephrologists, nurses, pharmacists, social workers and dietitians. Additionally, medical costs are much higher with increasing severity of CKD. As such, timely referral of CKD patients to the multidisciplinary renal unit is crucial to delaying kidney disease progression, reducing the initial cost of care even before the commencement of dialysis and decreasing the chance of kidney transplantation, comorbidities, and mortality [2].

In the renal unit, nutritional care is recommended for all stages of CKD since it aims to improve the nutritional status of patients, slowing the progression to dialysis by delivering tailored educational and dietary advice [3]. Despite the importance of nutritional management in CKD, the amount of published data regarding the effects of diet on CKD is lacking, both in Arab countries and worldwide. However, the few existing studies agree on the positive impact of personalized nutritional support to CKD patients.

In fact, it has been highlighted that an individualized diet, compared to the control, produces a significantly beneficial effect in terms of quality of life [4,5,6], mortality [7], urinary protein, blood pressure, urinary sodium, and total cholesterol levels [8,9], with a smaller decline in the estimated glomerular filtration rate (eGFR) [10]. On the other hand, few studies reported that the relative increase in the eGFR between groups was not significant [8,11].

Given this background, the aim of this study was to determine whether nutritional therapy has a role in slowing the progression of kidney failure in a cohort of Bahraini outpatients, through the assessment of the eGFR as the main outcome. The secondary objective of this study was assessing the effectiveness of dietary therapy by the evaluation of different biomarkers

## 2. Materials and Methods

This study is a retrospective, single-center cohort study. The protocol was approved by the health/social care community care research in the Ministry of Health (MOH) in Bahrain and the ethics committee of the Department of Biology, University of Bahrain, Bahrain. The university trial registration number is MAD/127/2020. 

### 2.1. Setting and Participants

This study was conducted at Salmaniya Medical Complex (SMC) in Manama, Bahrain. All participants gave their informed consent prior to their inclusion in this study in accordance with the Declaration of Helsinki. The inclusion criteria were patients older than 18 years of age with confirmed diagnosis of CKD. Patients who dropped out from the protocol, or with a single or transplanted kidney, or diagnosed with acute kidney injury, kidney stones, liver disease, polycystic kidney disease, nephrotic syndrome or cancer were excluded from this study. After applying the inclusion/exclusion criteria, a total of 265 patients were calculated with 147 patients in the non-nutritional intervention control group (non-NIG) and 118 patients in the nutritional intervention group (NIG) by using t-statistic and non-centrality parameter and taking into consideration a power rate of 80% [12,13]. The sample was based on the study by De Waal et al. [10] considering a primary outcome (eGFR) difference of 9.6 mL/min. 

### 2.2. Data Collection/Procedures

Baseline and follow-up data were obtained from the electronic medical records system of outpatients visiting SMC from 1 January 2016 to 31 December 2019. Patients were allocated into two groups, the NIG and the non-NIG that did not receive any dietary support. Data collected at baseline included age, gender, anthropometric and hematochemical data, drugs used, comorbidities and follow-up period. Laboratory data included values for hemoglobin, g/dL; serum albumin g/L; eGFR, mL/min/1.73 m^2^; blood urea nitrogen (BUN), mmol/L; creatinine, µ mol/L; urinary albumin/creatinine ratio (UACR), mg/mmol; microalbuminuria, mg/L; calcium, mmol/L; serum phosphorus, mmol/L; potassium, mmol/L; sodium, mmol/L; parathyroid hormone (PTH), pmol/L; uric acid, µmol/L; total cholesterol, mmol/L; low-density lipoprotein (LDL), mmol/L; high-density lipoprotein (HDL), mmol/L; triglyceride, mmol/L; fasting blood glucose (FBG), mmol/L; and glycated hemoglobin (HbA1c), mmol/mol. Baseline time was considered as the date of CKD diagnosis by the nephrologist for the non-NIG subjects and the date of the first visit to the renal dietitian for the NIG patients. The National kidney Foundation–Kidney Disease Outcomes Quality Initiative (NKF-KDOQI) clinical practice guidelines were applied for patients with an eGFR lower than 30 mL/min/1.73 m^2^ involving routine monitoring of nutritional status at 1–3 months intervals and a follow-up every 6–12 months for patients with an eGFR of 30–60 mL/min/1.73 m^2^ [14].

### 2.3. Nutritional Intervention

Individualized nutrition care was provided for patients in the NIG, by a trained renal dietitian, according to their level of kidney function, electrolytes abnormalities, comorbidities and nutrition status. Nutrition care plans focused on providing face-to-face counseling and meeting adequate energy and protein requirements with low sodium and modified potassium and phosphorus intakes according to the Academy of Nutrition and Dietetics’ guidelines for adults with CKD. Specifically, the nutritional recommendations for adults with chronic kidney disease stages 3 to 4 were set for protein ≤0.8 g/kg/day, increase plant source, salt <2.3 g/day (<5 g/day of NaCl), potassium individualize to keep the serum potassium within a normal range, calcium 1.5 g/day from both dietary and medication sources, phosphorus 0.8 to 1 g/day or individualize to keep the value within a normal range, increase vegetable source and avoid processed foods as much as possible. Carbohydrate/fat from 30 to 35 kcal/kg/day; <30% of total calories from fat and <10% of total fat from saturated fat; and fiber from 25 to 38 g/day (KDOQI guidelines) [3].

On the other hand, patients in the non-NIG were not visited by the renal dietitian but were given generic nutritional information containing dietary advice that is typically provided to CKD patients by nephrologists. 

The participants’ recommendations were followed up by the dietitian monthly through the FFQ. 

### 2.4. Statistical Analysis

All the data were analyzed by using Statistical Package for the Social Sciences (SPSS) package version 23.0, Armonk, NY, USA). Continuous variable analyses were reported as the mean ± standard deviation. The data collected from both groups were compared within and between the groups by using analysis of covariance (ANCOVA). Differences were considered as statistically significant with *p* value < 0.05.

## 3. Results

Of the 1213 patients admitted in the out-patient nephrology clinic during the specified period, 271 patients were identified and included as the study sample. The excluded patients (77.66%) were those who were below 18 years, lacking laboratory data, lost during follow-up, renal transplant recipients, and diagnosed with any of the following: acute kidney injury, kidney stones, liver disease, polycystic kidney disease, nephrotic syndrome, or cancer. The enrolled patients were divided into two groups: the intervention group (NIG), where 121 patients received counselling from a renal dietitian, and the control group (non-NIG), where 150 patients did not receive consultation (Figure 1).

Many of the enrolled patients had hypertension (87.8%) and diabetes (79.7%), but lower than one-fifth (19.3%) had had heart disease. In terms of the follow-up period, 83.6% of the patients received a follow-up service within 0–6 months, while only 16.3% received a follow-up after 7–15 months after the baseline evaluation (Table 1).

### 3.1. Baseline Characteristics

Most of the patients were males (60.1%) and elderly, with a mean age of 66.5 ± 13.6 years. Nonetheless, the NIG patients were relatively young compared to the non-NIG patients (64.7 years ± 12.0 versus 67.9 years ± 14.6), and this difference was nearly statistically significant (*p* = 0.051) (Table 2).

In terms of baseline characteristics, there were no statistically significant differences between the two groups (Table 2) with the exception of the eGFR (NIG: 29.18 mL/min/1.73 m^2^ ± 13.213; non-NIG 32.87 mL/min/1.73 m^2^ ± 16.166; *p*-value = 0.044), and albumin levels (NIG: 41.56 ± 3.89; non-NIG: 39.69 g/L ± 4.48; *p*-value < 0.001).

### 3.2. Intervention Effect within Group (Pre-Post Intervention)

Anthropometric data in the NIG showed a significant decrease in body weight and body mass index (BMI) following the intervention (∆ = −1.8402 kg (−2.6069; −1.0736) and BMI ∆ = −0.6904 kg/m^2^ (−1.0039; −0.3768); *p* = 0.000, respectively), while in the control, the non-NIG, the change was not significant (Table 3).

Moreover, the change in the eGFR from baseline to follow-up in the NIG significantly increased (∆ = 5.165 * mL/min/1.73 m^2^ (4.04; 6.28); *p* = 0.000). Conversely, the control, the non-NIG, showed a significant decrease in the eGFR (∆ = −2.85 mL/min/1.73 m^2^ (−4.02; −1.67); *p* = 0.000). As shown in Table 3, nutritional intervention produced a beneficial positive impact on lipid profile, with a statistically significant decline in both total cholesterol and triglyceride levels in the NIG patients (∆ = −0.17 * mmol/L (−0.30; −0.05); *p* = 0.006 and ∆ = −0.15 * mmol/L (−0.28; −0.01); *p* = 0.026, respectively). However, the decrease in LDL level was not significant. In the non-NIG patients, the levels of total cholesterol, LDL, HDL, and triglycerides remained stable. Furthermore, FBG levels were significantly improved among the NIG patients (∆ = − 0.70 * mmol/L (−1.27; −0.13); *p* = 0.016), while the HbA1C values remained unchanged in both groups at follow-up. Similar statistically significant changes in the NIG were found in levels of BUN and creatinine, uric acid and phosphorous, with all showing a decrease from the baseline.

### 3.3. Between-Groups Effects

Results revealed statistically significant effects when comparing nutritional intervention with the non-NIG. As the control, the non-NIG, had no changes within the group, apart from the eGFR, the differences between groups were similar to the previously reported changes in the NIG. In fact, statistically significant Δ changes (NIG–non-NIG) were noted for body weight (∆ = −1.81 * kg; *p* = 0.022), BMI (∆ = −0.67 * kg/m^2^; *p* = 0.021), the eGFR (∆ = 8.01 * mL/min/1.73 m^2^; *p* = 0.000), BUN (∆ = −2.84 * mmol/L, *p* = 0.000), creatinine (Δ = −23.58 µmol/L, *p* = 0.000), total cholesterol (Δ = −0.31 * mmol/L; *p* = 0.0319, and uric acid (∆ = −35.25 * µmol/L; *p* = 0.004). However, there were no significant differences between the groups following the intervention for albumin, sodium, potassium, calcium, phosphorus, PTH, and hemoglobin (Table 4).

## 4. Discussion

The present study revealed that a personalized nutritional intervention, provided by a renal dietitian, plays a significant role in slowing renal disease progression, as long as it is evaluated by the eGFR. Furthermore, it should be underlined that the eGFR was not only higher compared to the control at the end of the intervention, but also increased from the baseline in the NIG. As previously reported by several authors [3,15,16,17,18], the results demonstrate the potential of a CKD diet to facilitate weight management by reducing body weight and BMI in a cohort of mostly obese patients, together with the amelioration of both diabetes and CKD, thereby inducing a beneficial effect on several fronts. Indeed, a retrospective US cohort study showed that patients receiving nutrition therapy guidance had a lower decline in the eGFR and lower risk of starting dialysis than those not receiving nutrition therapy [10]. Another interventional study in Italy involving 16 CKD patients with stages 3–4 revealed the eGFR remaining stable in patients who received a personalized nutritional counselling session with a renal dietitian [8]. A more recent retrospective study showed that nutrition counselling provided to pre-dialysis patients was associated with delay time to dialysis [19].

Our results from this study showed that nutritional intervention decreased BUN and serum creatinine levels in CKD patients. This could be attributed to improved kidney function which might be related to the protein-restricted diet usually prescribed to CKD patients. Excessive protein intake is, in fact, associated with increased kidney workload and renal hyperfiltration. Additionally, protein restriction may reduce uremic symptoms and delay the time of dialysis [3]. A previous US study reported a significant reduction in BUN levels in the nutritional intervention group, while serum creatinine levels remained stable [8]. Similarly, Cupisti et al. [11] revealed a reduction in serum BUN levels in CKD patients at stages 4 and 5 after nutrition intervention comparted to those in the control group.

As previously reported, nutrition intervention had a significant effect on reducing both body weight and BMI in terms of changes from baseline and between-group differences. It is important to underline that, since the mean BMI was above 30 kg/m^2^, the dietician had to set a proper hypocaloric diet for nearly all the patients. During this study, the body weight of the non-NIG patients remained constant, leading to hypothesize that weight management was not part of traditional nephrological indications, or at the least failed due to the lack of providing personalized advice.

In addition, we found that nutritional intervention in pre-dialysis patients was associated with a decrease in FBG, total cholesterol and triglyceride levels, and a slight decrease in LDL level, which was not statistically significant. This finding could definitely be attributed to the nutrition intervention guidelines for CKD patients that recommends modification of the type and amount of fat intake including replacing saturated fats with unsaturated fats and focusing and increasing dietary intake of fiber along with physical activity levels [18,20]. Moreover, the most recommended dietary pattern for non-dialysis patients is the Mediterranean-style diet [3]. In fact, similar effects were reported after a 3-month intervention period in a group of CKD patients following a Mediterranean diet, both in terms of total cholesterol and triglycerides levels [21]. Another study found that pre-dialysis patients who received dietetic consultation for more than 12 months had lower total cholesterol levels when starting dialysis [7]. In contrary to our findings, Cupisti et al. [11] reported that serum lipid profile (total cholesterol, LDL, HDL, and triglycerides) was roughly the same in both the treatment group and the control group.

Nutritional intervention was also associated with a decrease in uric acid levels. This is probably due to the modification of the CKD diet, which aims to control blood sugar and blood pressure levels and restrict protein intake through individualized nutrition assessment, care planning and nutrition education, all of which play a role in decreasing the risk of developing hyperuricemia [22]. Furthermore, the benefits of pre-dialysis nutritional intervention involve the decrease in mean levels of serum phosphorus. Protein restriction may have been beneficial in decreasing phosphorus levels in CKD patients. Another possible explanation could be that individualized nutritional counselling provided by a dietitian for CKD patients with hyperphosphatemia involves the restriction of phosphorus intake by providing information on the main dietary sources of phosphorus and the difference between organic (animal and plant) and inorganic sources (additives in processed foods) of phosphorus [23]. A similar outcome was reported in CKD patients after 16 months of receiving personalized nutritional intervention by a renal dietitian [8] as well as in a case–control study setting [11].

Among the most crucial aspects of the CKD diet is its effect on albumin levels. This study showed that a protein-restricted diet is associated with a slight reduction in albumin levels in the NIG patients; however, it was not statistically significant. It has been previously claimed that a hypoproteic diet does not severely affect albumin levels and tends to remain unchanged even after long periods of treatment [3,10]. There are also a couple of studies in the literature showing albumin levels increasing following nutritional support in CKD patients, probably due to the less severe hyporoteic diet [7,11].

Finally, the results from this study revealed that sodium, potassium, and calcium levels mostly remained within the reference ranges following the nutrition intervention. The data concerning potassium levels may have been the result of adherence to the CKD nutrition guidelines that recommends against restricting potassium intake unless patients are diagnosed with hyperkalemia. Considering the fact that the mean baseline levels of potassium in the NIG being within the normal ranges rendered the restriction unnecessary [16].

It worth nothing that CKD stage 4a+ in this investigated population in Bahrain is 52% and this is in line with data reported in the US (59.1%) [24].

### Study Strengths and Limitations

In this study, we present for the first time in the Arab world clear evidence of the superiority of a personalized nutritional intervention, combined with traditional nephrological assistance, in the treatment of CKD in outpatient subjects, thereby creating favorable health outcomes in terms of weight management, metabolic response, and dialysis postponement. The results of this study may assist regional and local authorities and governments in reducing health care costs related to dialysis treatment by paying more attention to preventive dietary measures. The limitations of this study were such that the baseline characteristics of the NIG and the non-NIG were not completely comparable, especially for the primary outcome: the eGFR due to the lack of randomization. However, this bias may be overcome by the extreme difference in terms of the eGFR between groups at the end of the intervention. Nonetheless, we believe that the evidence presented here provides support for additional research that involves two similar cohorts with proper randomization. Close monitoring of body composition with the use of bioimpedance or DXA for assessing body composition, muscle mass, strength, physical function were not provided in this study and should be presented as shortcomings of this study.

## Figures and Tables

**Figure 1 geriatrics-07-00083-f001:**
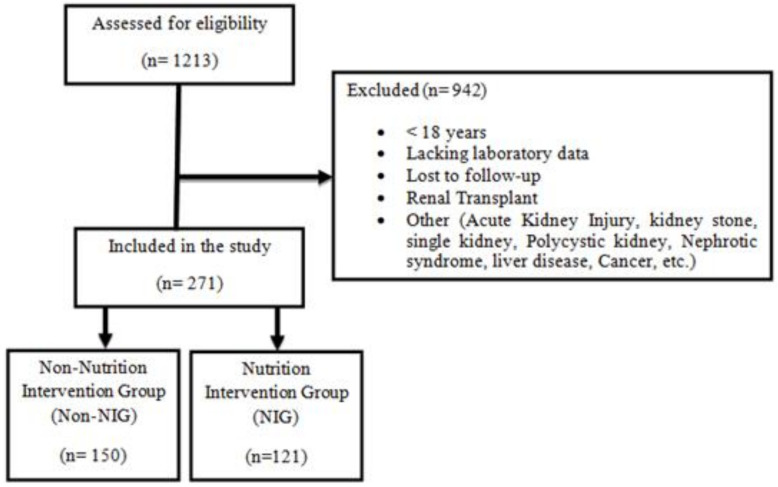
Flowchart of patients included and excluded in this study.

**Table 1 geriatrics-07-00083-t001:** Demographic data, co-morbidities, and follow-up period characteristics of the sample.

Variables	*n*	%
Sex		
Male	163	60.1
Female	108	39.9
**Co-morbidities**		
Diabetes	216	79.7
Hypertension	238	87.8
Heart disease	52	19.3
**Follow-up period**		
0–3 Months	97	41.8
4–6 Months	97	41.8
7–9 Months	30	12.9
10–15 Months	8	3.4
**CKD stages on the eGFR**		
60–45 mL/min	65	24
30–44.9 mL/min	65	24
<30 mL/min	141	52

**Table 2 geriatrics-07-00083-t002:** Characteristic of the sample.

Variable	Non-Nutrition Intervention Group (Non-NIG) (*n* = 150) 59 F, M 91	Nutrition Intervention Group (NIG) (*n* = 121) 49 F, 72 M	Total Sample (*n* = 271; Non-NIG = 150; NIG = 121)	*p*-Value between Groups
**Age** year.	67.95 ± 14.62	64.71 ± 12.01	66.51 ± 13.58	0.051
**Weight** kg	79.66 ± 17.03	83.06 ± 16.99	81.86 ± 17.03	0.235
**BMI** kg/m^2^	30.18 ± 6.28	31.55 ± 7.22	31.06 ± 6.91	0.256
**Hemoglobin** g/dL	11.04 ± 1.86	11.33 ± 1.71	11.17 ± 1.80	0.202
**eGFR** mL/min/1.73 m^2^	32.87 ± 16.16	29.18 ± 13.21	31.21 ± 14.99	0.044 *
**T. cholesterol** mmol/L	4.01 ± 1.17	4.03 ± 0.95	4.02 ± 1.08	0.886
**HDL** mmol/L	1.13 ± 0.81	1.04 ± 0.32	1.09 ± 0.64	0.237
**LDL** mmol/L	2.20 ± 1.11	2.22 ± 0.93	2.21 ± 1.03	0.874
**Albumin** g/L	39.69 ± 4.48	41.56 ± 3.89	40.55 ± 4.31	*p* < 0.001 *
**FBG** mmol/L	6.85 ± 3.13	7.17 ± 3.70	7.00 ± 3.40	0.465
**HbA1C** mmol/mol	55.07 ± 19.65	53.11 ± 17.22	54.14 ± 18.52	0.414
**BUN** mmol/L	15.10 ± 7.66	16.88 ± 8.17	15.88 ± 7.92	0.079
**Creatinine** µmol/L	233.12 ± 310.93	224.01 ± 120.58	229.06 ± 244.68	0.762
**Microalbumin** mg/g	339.75 ± 634.34	371.48 ± 555.33	355.15 ± 595.32	0.757
**UACR** mg/g	79.91 ± 172.51	98.75 ± 170.77	90.61 ± 171.24	0.489
**PTH** pmol/L	19.96 ± 20.69	20.00 ± 19.42	19.98 ± 20.07	0.990
**Uric acid** µmol/L	438.00 ± 103.49	444.89 ± 123.91	441.12 ± 113.02	0.628
**Sodium** mol/L	149.02 ± 107.22	140.25 ± 2.90	145.15 ± 80.15	0.377
**Potassium** mol/L	4.95 ± 0.63	5.02 ± 0.70	4.98 ± 0.66	0.387
**Calcium** mol/L	2.42 ± 2.03	2.47 ± 1.95	2.44 ± 1.99	0.830
**Phosphorus** mol/L	1.51 ± 1.47	1.33 ± 0.31	1.43 ± 1.10	0.211
**Follow-up period** months	6 ± 2	6 ± 4	6 ± 3	0.850

Abbreviation: BMI, body mass index; T. cholesterol, total cholesterol; HDL, high-density lipoprotein; LDL, low-density lipoprotein; FBS, fasting blood sugar; HbA1c, hemoglobin A1c; BUN, blood urea nitrogen; UACR, urine albumin-to-creatinine ratio; PTH, parathyroid hormone. *: *p* < 0.05.

**Table 3 geriatrics-07-00083-t003:** Within-group effects.

Variable	Non-Nutrition Intervention (Non-NIG), within-Group Δ Change (SD)	*p*-Value within Group	Nutrition Intervention (NIG), within-Group Δ Change (SD)	*p*-Value within Group
**Weight** kg	−0.02 (−1.43; 1.38)	0.970	−1.84 * (−2.60; −1.07)	*p* < 0.001
**BMI** kg/m^2^	−0.01 (−0.61; 0.58)	0.960	−0.69 *(−1.00; −0.37)	*p* < 0.001
**Hb** g/dL	−0.02 (−0.21; 0.17)	0.837	−0.00 (−0.21; 0.20)	0.951
**eGFR** mL/min/1.73 m^2^	−2.85 * (−4.02; −1.67)	0.000	5.16 * (4.04; 6.28)	*p* < 0.001
**Cholesterol** mmol/L	0.13 (−0.05; 0.31)	0.155	−0.17 * (−0.30; −0.05)	0.006
**HDL** mmol/L	−0.06 (−0.21; 0.07)	0.341	0.02 (−0.01; 0.05)	0.219
**LDL** mmol/L	0.09 (−0.11; 0.30)	0.382	−0.15 (−0.31; 0.00)	0.060
**Triglyceride** mmol/L	0.06 (−0.09; 0.22)	0.411	−0.15 * (−0.28; −0.01)	0.026
**Albumin** g/L	0.26 (−0.38; 0.91)	0.424	−0.29 (−0.77; 0.18)	0.229
**F.B.S** mmol/L	0.22 (−0.44; 0.89)	0.513	−0.70 * (−1.27; −0.13)	0.016
**HbA1c** mmol/mol	0.35 (−2.28;3.00)	0.789	1.10 (−1.58; 3.79)	0.419
**BUN** mmol/L	0.65 (−0.02; 1.33)	0.059	−2.19 * (−2.96; −1.42)	*p* < 0.001
**Creatinine** µmol/L	−1.72 (−46.83; 43.39)	0.940	−25.31 * (−30.63; −19.98)	*p* < 0.001
**Microalbumin** mg/g	18.15 (−164.73; 201.04)	0.842	−104.21 (−249.97; 41.53)	0.157
**UACR** mg/g	20.45 (−24.56; 65.47)	0.365	−9.13 (−35.21; 16.94)	0.488
**PTH** pmol/L	2.87 (−0.52; 6.27)	0.097	−0.83 (−3.34; 1.66)	0.508
**Uric acid** µmol/L	6.90 (−11.44; 25.25)	0.458	−28.35 * (−48.55; −8.15)	0.006
**Sodium** mmol/L	−9.36 (−27.24; 8.50)	0.302	11.94 (−11.37; 35.27)	0.312
**Potassium** mmol/L	0.04 (−0.05; 0.14)	0.413	0.00 (−0.11; 0.11)	1.000
**Calcium** mmol/L	−0.14 (−0.51; 0.23)	0.453	−0.20 (−0.58; 0.16)	0.275
**Phosphorus** mmol/L	−0.14 (−0.46; 0.18)	0.393	−0.05 * (−0.10; −0.00)	0.025

Abbreviation: BMI, body mass index; T. cholesterol, total cholesterol; HDL, high-density lipoprotein; LDL, low-density lipoprotein; FBS, fasting blood sugar; HbA1c, hemoglobin A1c; BUN, blood urea nitrogen; UACR, urine albumin-to-creatinine ratio; PTH, parathyroid hormone. * *p* < 0.001. * Data adjusted for age, gender and length of follow up.

**Table 4 geriatrics-07-00083-t004:** Mean change in the nutrition intervention group (NIG) (effect of) minus the effect of the non-nutrition intervention group (non-NIG) from baseline to follow-up. * *p* < 0.001. * Data adjusted for age, gender and length of follow up. Abbreviation: BMI, body mass index; T. cholesterol, total cholesterol; HDL, high-density lipoprotein; LDL, low-density lipoprotein; FBS, fasting blood sugar; HbA1c, hemoglobin A1c; BUN, blood urea nitrogen; UACR, urine albumin-to-creatinine ratio; PTH, parathyroid hormone.

Variable	Nutrition Intervention Group NIG (Effect of) Minus the Effect of Non-Intervention Group (Non-NIG) Δ Change	*p*-Value
**Weight** kg	−1.81 *	0.022
**BMI** kg/m^2^	−0.67 *	0.021
**Hemoglobin** g/dL	0.01	0.534
**eGFR** mL/min/1.73 m^2^	8.01 *	0.000
**T. Cholesterol** mmol/L	−0.31 *	0.031
**HDL** mmol/L	0.71	0.250
**LDL** mmol/L	−0.24	0.151
**Triglycerides** mmol/L	0.22	0.100
**Albumin** g/L	−0.55	0.465
**FBS** mmol/L	−0.92	0.086
**HbA1C** mmol/mol	0.74	0.484
**BUN** mmol/L	−2.84 *	*p* < 0.001
**Creatinine** µmol/L	−23.58 *	*p* < 0.001
**Microalbumin** mg/g	−122.36	0.499
**UACR** mg/g	−29.58	0.488
**PTH** pmol/L	−3.70	0.098
**Uric acid** µmol/L	−35.25 *	0.004
**Sodium** mol/L	21.31	0.575
**Potassium** mol/L	−0.04	0.955
**Calcium** mol/L	−0.06	0.289
**Phosphorus** mol/L	0.08	0.856

## Data Availability

Not applicable.

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
