# Peer review of "Nutritional Intervention Effectiveness on Slowing Time to Dialysis in Elderly Patients with Chronic Kidney Disease—A Retrospective Cohort Study"

_geriatrics, 2022, doi:10.3390/geriatrics7040083_

Round 1

Reviewer 1 Report

Nutritional Intervention Effectiveness on Slowing-Dialysis in Elderly Patients with Chronic Kidney Disease. A Retrospective Cohort Study

Overall, it is potentially interesting study, especially for readers from Arab countries. The manuscript is generally well-written.

Comments

It is not clear, how the participants were allocated to NIG and non-NIG groups. NIG group was younger, probably in better nutritional status (higher albumin level). The sex composition of both groups should also be provided.

The time of observation period (mean or median) should be shown.

The nutritional intervention should be described in more detail. The mode of controlling the participants how they follow the recommendations should be clearly presented.

Protein-restricted diet may have beneficial effect in patients with chronic kidney disease (not all – what was the percentage of subjects with nephrotic syndrome?).

On the other hand, older patients should be provided with an adequate protein intake to sustain muscle mass and function. Those data (e.g. bioimpedance body composition, muscle strength, physical function) were not provided and should be presented as shortcomings of the study.

Author Response

REV 1

1It is not clear, how the participants were allocated to NIG and non-NIG groups. NIG group was younger, probably in better nutritional status (higher albumin level). The sex composition of both groups should also be provided.

Answer: the allocation was random. Unfortunately. NIG group was younger (yes but not statistically significant),

You mentioned that probably in better nutritional status (higher albumin level), despite the mean level was 39 versus 41 and it does not impact of the nutritional status.

2The time of observation period (mean or median) should be shown.

Answer: we added this variable in table 2

3The nutritional intervention should be described in more detail.

Answer. We described in detail the nutritional intervention

4The mode of controlling the participants how they follow the recommendations should be clearly presented.

Answer. The participant’s recommendations have been followed up by the dietitian at monthly level by the administration of the FFQ.

5Protein-restricted diet may have beneficial effect in patients with chronic kidney disease (not all – what was the percentage of subjects with nephrotic syndrome?)

Answer: as we described inclusion criteria, we excluded patients with nephrotic syndrome, liver disease, Cancer, etc.).

6On the other hand, older patients should be provided with an adequate protein intake to sustain muscle mass and function. Those data (e.g. bioimpedance body composition, muscle strength, physical function) were not provided and should be presented as shortcomings of the study.

Answer. An ad hoc  paragraph into the conclusion has been written following your advise.

Reviewer 2 Report

In this study, the authors evaluated the impact of nutritional intervention on kidney disease progression in a cohort of patients with CKD. The manuscript is well written, and study results seem significant as Nutritional intervention led to improved eGFR and other biomarkers over time; however, I have many concerns to be addressed:

-Methodology:

·       It is not clear how allocation of patients to the 2 groups was performed (retrospectively determined?). Be more accurate in reporting this part of methodology in the first paragraph (instead of lines 91-92). From lines 82-84, it seems that allocation of patients was determined after selection of patients included in the study, which would contrast with the retrospective design of the study.

·       Inclusion criteria: you included patients with a confirmed diagnosis of CKD, but I am surprised by the very low mean eGFR value in the two groups, especially in the non-NIG group (when the lab evaluation was performed at the time of CKD diagnosis); I would expect a higher baseline value for those with a new diagnosis of CKD. I am wondering whether eGFR is normally distributed or not in the 2 subgroups. Please investigate it, and also report the prevalence of CKD stages 60-45, 30-44.9 and < 30 ml/min.

If data are confirmed, it might mean that CKD diagnosis is often delayed in this setting; add a few lines to comment this issue, comparing with other studies (overall prevalence of CKD stages according to KDIGO classification in Bahrain vs other countries).

·       I am concerned about the different follow-up times in patients included in the study. At lines 104-106 you specified that patients underwent monitoring at 1-3 months when eGFR < 30 ml/min. However, despite the mean eGFR was < 30 ml/min, it seems that no patient underwent more than 1 follow-up visit (see table 1). Additionally, were patients visited more than 1 time during the follow-up characterized by a better improvement   in eGFR compared to those visited 1 time? Moreover, the first visit was always at time 0? (e.g for patients with the follow-up visit at 4-6 months, or those for 7-9 months?). Which was the mean follow-up time?

·       Were lab values collected at the first visit only or were they repeated during follow-up?

Statistical analysis

·       Were all continuous variables normally distributed? How did you assess normality of their distributions? (Specify in the section of statistical analysis). Furthermore, consider the opportunity of stratifying analyses by CKD stages, as it might add more significance to study results (were there differences in study results depending on baseline CKD stage?).

·       Which FU time have you used for ANCOVA ? Was analysis adjusted for age or other covariates? In the case of variable FU time, did FU duration affect study results?

Minor changes:

·       Table 1: add eGFR and CKD stages.

·       How many patients progressed to dialysis in the study?

·       All the manuscript: change P=0.000 with P<0.001.

·       Line 43: change “causes” with “cause”.

·       Line 43-44: is this trend related to the last decade? Add timeline and reference.

·       Line 45: change “require” with “requires”.

·       Line 66: “through the assessment of eGFR level as the main outcome”. Improve definition of the main outcome of the study (eGFR variation at the end of the FU?).

·       Line 115: add reference.

·       Add a Supplementary Box including nutritional tips for the 2 groups (energy intake, micro and macronutrient intake).

·       Add number of patients excluded for each criterion to the flow-chart.

·       Add comorbidities to table 2.

Author Response

REV 2

Dear reviewer, thanks a lot for your efforts and time dedicated to our paper. We really appreciated all your time and we did our best for improving it in line with our vision and best efforts. 

-Methodology:

  • It is not clear how allocation of patients to the 2 groups was performed (retrospectively determined?). Be more accurate in reporting this part of methodology in the first paragraph (instead of lines 91-92). From lines 82-84, it seems that allocation of patients was determined after selection of patients included in the study, which would contrast with the retrospective design of the study.

Answer: the allocation was random and done retrospectively at the beginning of this study.

Being a retrospective study, the allocation of patients was determined before (and not after) the selection of patients included in the study.

Inclusion criteria: you included patients with a confirmed diagnosis of CKD, but I am surprised by the very low mean eGFR value in the two groups, especially in the non-NIG group (when the lab evaluation was performed at the time of CKD diagnosis); I would expect a higher baseline value for those with a new diagnosis of CKD. I am wondering whether eGFR is normally distributed or not in the 2 subgroups.

Answer: we checked in pre analysis and the eGFR is normally distributed in the 2 subgroups.

investigate it, and also report the prevalence of CKD stages 60-45, 30-44.9 and < 30 ml/min.

Answer: we made this analysis and included the data in table 1. 52% is < 30 ml/min.We compared this data with US data

If data are confirmed, it might mean that CKD diagnosis is often delayed in this setting; add a few lines to comment this issue, comparing with other studies (overall prevalence of CKD stages according to KDIGO classification in Bahrain vs other countries).

Answer: We added few line discussing this issue as you requested. We compared this data with US data

I am concerned about the different follow-up times in patients included in the study. At lines 104-106 you specified that patients underwent monitoring at 1-3 months when eGFR < 30 ml/min. However, despite the mean eGFR was < 30 ml/min, it seems that no patient underwent more than 1 follow-up visit (see table 1).

Answer: yes, patients had only one follow up. Patients had a monitor of the situation, not a proper follow up with all parameter evaluated. But only egfr. The value egfr was not recorded as follow up.

Additionally, were patients visited more than 1 time during the follow-up characterized by a better improvement   in eGFR compared to those visited 1 time?

Answer. We did not establish more than a follow up. It was just an informal meeting with the MD in order to monitor any side effect. Data were not included into the protocol or data collection

Moreover, the first visit was always at time 0? (e.g for patients with the follow-up visit at 4-6 months, or those for 7-9 months?). Which was the mean follow-up time?

Answer: also as requested by the reviewer 2, we included the mean and standard deviation of the follow up for each group

Were lab values collected at the first visit only or were they repeated during follow-up?

Answer: lab values were collected at baseline and during the only one follow up (end of the protocol)

Statistical analysis Were all continuous variables normally distributed? How did you assess normality of their distributions? (Specify in the section of statistical analysis).

Answer The Kolmogorov-Smirnov test was used to identify the normal distribution of the data. Data were normal distributed at baseline

Was analysis adjusted for age or other covariates?

Answer: Analysis have been adjusted for gender, age and length of follow up

Minor changes

ANSWER All minor comments have been addressed:

  • Table 1: add eGFR and CKD stages.
  • How many patients progressed to dialysis in the study? Data
  • All the manuscript: change P=0.000 with P<0.001.
  • Line 43: change “causes” with “cause”.
  • Line 43-44: is this trend related to the last decade? Add timeline and reference.
  • Line 45: change “require” with “requires”.
  • Add a Supplementary Box including nutritional tips for the 2 groups (energy intake, micro and macronutrient intake).

Round 2

Reviewer 1 Report

Line 303, check:

CKD stage 4a+

Reviewer 2 Report

The authors addressed all my comments. The paper is now suitable for publication.